# Compromised Barrier Function in Human Induced Pluripotent Stem-Cell-Derived Retinal Pigment Epithelial Cells from Type 2 Diabetic Patients

**DOI:** 10.3390/ijms20153773

**Published:** 2019-08-01

**Authors:** Mostafa Kiamehr, Alexa Klettner, Elisabeth Richert, Ali Koskela, Arto Koistinen, Heli Skottman, Kai Kaarniranta, Katriina Aalto-Setälä, Kati Juuti-Uusitalo

**Affiliations:** 1Faculty of Medicine and Health Technology, Tampere University, 33014 Tampere, Finland; 2Department of Ophthalmology, University of Kiel, University Medical Center, 24105 Kiel, Germany; 3Department of Ophthalmology, Institute of Clinical Medicine, University of Eastern Finland, 70210 Kuopio, Finland; 4SIB Labs, University of Eastern Finland, 70210 Kuopio, Finland; 5Department of Ophthalmology, Kuopio University Hospital, 70210 Kuopio, Finland; 6Department of Molecular Genetics, University of Lodz, 90-136 Lodz, Poland; 7Heart Hospital, Tampere University Hospital, 33521 Tampere, Finland

**Keywords:** retinal pigment epithelial cells, type 2 diabetes, diabetic retinopathy (DR), induced pluripotent stem cells (hiPSC), barrier function, autophagy, matrix metalloproteinase

## Abstract

In diabetic patients, high blood glucose induces alterations in retinal function and can lead to visual impairment due to diabetic retinopathy. In immortalized retinal pigment epithelial (RPE) cultures, high glucose concentrations are shown to lead to impairment in epithelial barrier properties. For the first time, the induced pluripotent stem-cell-derived retinal pigment epithelium (hiPSC-RPE) cell lines derived from type 2 diabetics and healthy control patients were utilized to assess the effects of glucose concentration on the cellular functionality. We show that both type 2 diabetic and healthy control hiPSC-RPE lines differentiate and mature well, both in high and normal glucose concentrations, express RPE specific genes, secrete pigment epithelium derived factor, and form a polarized cell layer. Here, type 2 diabetic hiPSC-RPE cells had a decreased barrier function compared to controls. Added insulin increased the epithelial cell layer tightness in normal glucose concentrations, and the effect was more evident in type 2 diabetics than in healthy control hiPSC-RPE cells. In addition, the preliminary functionality assessments showed that type 2 diabetic hiPSC-RPE cells had attenuated autophagy detected via ubiquitin-binding protein p62/Sequestosome-1 (p62/SQSTM1) accumulation, and lowered pro- matrix metalloproteinase 2 (proMMP2) as well as increased pro-MMP9 secretion. These results suggest that the cellular ability to tolerate stress is possibly decreased in type 2 diabetic RPE cells.

## 1. Introduction

Diabetic retinopathy (DR), the major complication in patients with diabetes, is a leading cause of preventable vision loss in working-aged people in industrialized nations [1,2,3]. DR is estimated to account for 15–17% of total blindness in Europe and USA [4]. Twenty years after the onset of diabetes, almost all (90%) patients with type 1 diabetes and half (50%) of patients with type 2 diabetes have some form of DR. One third of patients with type 2 diabetes have DR-type changes at the time of diagnosis [1,4]. 

DR can be divided into nonproliferative diabetic retinopathy (NPDR) and proliferative diabetic retinopathy (PDR). The NPDR is characterized by microaneurysms (i.e., dilation of capillaries, (Figure 1a), hard and soft exudates (Figure 1a), vessel dilatations and tortuosity, haemorrhages (Figure 1a), and intraretinal microvascular abnormalities (Figure 1b). 

NPDR can be classified according to mild (only microaneurysms), moderate (more than microaneurysms), and severe retinopathy (many complications in all retina sections) levels. The NPDR stages are primarily a consequent of increased hyperglycemic, oxidative stress, hypoxia, and inflammation conditions that may lead to the PDR, which is characterized by the growth of new, but fragile, retinal vessels [1]. The vascular leakage from these newly formed vessels can lead to swelling of the macula, and hence to the macular edema, which is the most common cause of blindness in diabetic patients [1,5]. Endothelial pathology in DR has been characterized in detail, but the effects in the outer retina, in retinal pigment epithelium (RPE), has drawn rather little attention [1,2,3,5,6,7,8]. 

RPE cells are a polarized monolayer of cells that form a tight and selective diffusion barrier between the choroid and the neural retina [1,2,3]. It has been widely accepted that the increased glucose concentration escalates leakiness of the outer retina [1,2,3]. The hyperglycemic stimuli have been shown to decrease the barrier function in diabetic rodents [9] and decrease the tightness of the RPE cell layer in immortalized ARPE-19 cell cultures [10,11]. The hyperglycemic stimuli has also been shown to increase extracellular matrix protein, such as collagen and laminin [12,13], extracellular matrix organizing protein, such as matrix metalloproteinases (MMP) [14,15] production, and the expression of pigment epithelial growth factor (PEDF) [16,17]. 

Pathogenesis of DR has been studied for decades by using rodents, dogs, pigs, and non-human primates [1], as well as retinal ex vivo cultures [18] and primary cells isolated from diabetic patients [15,16]. However, the limitation of these diabetic patient cell/tissue culture approaches is their inherent heterogeneity [1]. Therefore, a feasible option to achieve a disease-specific cell type is to use donor specific human induced pluripotent stem cells (hiPSCs) [1,19].

Autophagy is a host defense response to many environmental stresses observed in retinal diseases such as glycative stress, nutrient deprivation, hypoxia, or oxidative stress [20]. In autophagy, the autophagosome seals the content destined to degradation and fuses with a lysosome to degrade the cargo. In this process, the p62/SQSTM1 (sequestosome 1) protein, which also has a role in anti-oxidant cytoprotection via the NRF-2/ARE (nuclear erythroid 2-related factor 2/antioxidant response element) pathway, sort and bind the cargo for autophagic degradation and binds with LC3 (microtubule-associated protein 1A/1B-light chain) to form autophagosome and seal the cargo [21]. Both of these proteins are degraded along with the cargo and are therefore used for monitoring autophagy markers [22]. Similarly, increased oxidative stress, disturbed proteostasis, and accumulated toxic compounds trigger the progression from para-inflammation to chronic inflammation [20,23]. 

In previous studies, diabetic patient-derived hiPSCs have already been differentiated into cardiomyocytes [24] and insulin-producing cells [25], but according to our knowledge, there is no publications of differentiation of hiPSCs into RPE cells. In this study, we assessed the development of the barrier function of type 2 diabetic patient-derived hiPSC-RPEs assessed under different glucose concentrations in the presence or absence of added insulin. In addition, the functionality was preliminarily investigated under oxidative stress, and autophagic and cytokine stimuli.

## 2. Results

### 2.1. Pluripotency Assessment

Here, the hiPSC lines, three (UTA.08002.DMs, UTA.08203.DMs, and UTA.10802.EURCCs) from type 2 diabetics and two (UTA.10212.EURCCs, and UTA.10902.EURCCs) from controls used in this study were characterized in detail for their pluripotency. In all five hiPSC lines the virally transferred exogenous pluripotency genes (*OCT4*, *c-MYC*, *SOX2*, and *KLF*) were silenced (Figure 2a) and they all expressed the pluripotency markers of *OCT4*, *REX1*, *SOX2*, *NANOG*, and *c-MYC* at the gene level (Figure 2b). The pluripotency of iPSC lines was verified *in vitro* using embryoid body(EB) formation, from which we showed using PCR that EBs were expressing at least one marker from each of the three germ layers (endoderm, mesoderm, and ectoderm) (Figure 2c). Furthermore, the results from indirect immunofluorescence staining confirmed the expression of NANOG, OCT4, SOX2, SSEA4, TRA1-60, and TRA1-80 at the protein level (Figure 2d). During the time of the experiment, the karyotypes of all five iPSC lines were normal (Figure 2e).

### 2.2. Maturation Status of Cells Verified with Gene Expression and Morphology

The maturation of PREs was characterized using visual detection of pigmentation (Figure 3a–d). Although there were slight differences in the intensity of pigmentation, all characterized cell lines had a cobblestone morphology and pigmentation. In the representative immunofluorescent staining, all analyzed cell lines were positive for tight junction localizing ZO-1 (Figure 3e–h). The maturation was further assessed with transmission electron microscopy (TEM), in which all the characterized cell lines exhibited pigment granules as well as a thick and even brush border (Figure 3i–l). 

The differentiation and maturation status of hiPSC-RPEs derived from type 2 diabetics (UTA.08002.DMs, UTA.08203.DMs, and UTA.10802.EURCCs), and healthy control (UTA.10212.EURCCs and UTA.10902.EURCCs) patient hiPSC-RPEs was assessed using RT-PCR after five weeks of culture on polyethylene terephthalate (PET) inserts in normal (NG) or high glucose (HG) (Figure 3m). The pluripotency gene *Oct3/4*, a marker of undifferentiated hiPSCs, was quiescent, and *PAX6*, a marker of neuroectodermal and eye-specific lineage, was expressed in low levels. *Bestrophin* (*BEST1*), a RPE-specific gene, was expressed in all analyzed cell lines. The *RPE65*, another RPE-specific gene, was expressed in all cell lines. The *tyrosinase* gene, which is essential for melanin synthesis, was expressed in high levels in type 2 diabetic hiPSC-RPE line UTA.08002.DMs, and in healthy control hiPSC-RPE lines UTA.10212.EURCCs and UTA.10902.EURCCs, and low levels in type 2 diabetic hiPSC lines UTA.08203.DMs and UTA.10802.EURCCs (Figure 3m).

### 2.3. Barrier Properties in hiPSC-RPE Cells

When RPE cells mature, they form a tight, uniform, and polarized cellular monolayer. We followed the maturation of hiPSC-RPE cells derived from diabetic or healthy control individuals grown in different high or normal glucose concentrations in the presence or absence of added insulin over five weeks. The maturation of epithelial cell layer was evaluated by assessing the trans-epithelial electrical resistance (TEER). Both the type 2 diabetic and healthy control hiPCS-RPEs matured and the TEER increased during the follow-up period (Figure 4a,b). TEER in HG+ in type 2 diabetic cells was 313 Ω∙cm^2^ and in healthy control cells in HG+ was 208 Ω∙cm^2^. Statistical analysis verified that the difference was statistically significant (*p* = 0.03). There were statistically significant changes in TEER in NGM+, NG+, and NG− between the type 2 diabetic and healthy control cells, as illustrated in Figure 4c (NGM− *p* = 0.011, NG+ *p* = 0.017, and NG− *p* = 0.017). 

Then, we compared the TEER within cell groups when cultured in different glucose concentrations. Both type 2 diabetic and healthy control hiPSC-RPEs grown in HG+ had higher TEER than those grown in NGM−, and this difference was statistically significant (*p* = 0.01 and *p* = 0.016, respectively). In addition, there were statistically significant differences between conditions NGM− and NGM+, in type 2 diabetic (*p* = 0.006) and in healthy control (*p* = 0.041) cells.

The tightness of the epithelial cell layer was evaluated at the end of the cultivation period by assessing the cumulative permeability percentage of a small-molecular-weight fluorescent marker (FD4) through the hiPSC-RPE epithelial cell layer in an Ussing chamber system (Figure 5a–c). During the five weeks culture period, all hiPSC-RPE cultures had formed a tightly sealed cell layer as the cumulative permeability percentage of FD4 was close to 0 in both type 2 diabetic and healthy control hiPSC-RPEs (Figure 5a,b). When the endpoint at 240 min was plotted, it is seen that the type 2 diabetic hiPSC-RPE cultures were more permeable than healthy control hiPSC-RPEs (Figure 5c). The most statistically significant differences (*p* = 0.004) were found when the NGM+ condition was compared between type 2 diabetic (cumulative permeability % = −0.012) and healthy control (cumulative permeability % = −0.057) hiPSC-RPEs (Figure 5c).

### 2.4. Effects of Glucose Concentration on the Secretion of the RPE-Specific Growth Factor (PEDF) Secretion

Mature RPE cells normally synthetize and apically secrete PEDF growth factor. As PEDF is a marker of RPE cell wellbeing, and using this, we wanted to see whether different concentrations of glucose and/or insulin have an effect on PEDF secretion. During a 24 h collection time, all studied hiPSC-RPEs secreted a substantial amount (>500 ng/mL) of PEDF (Figure 6) from their growth area (0.3 cm^2^). The type 2 diabetic hiPSC-RPE lines secreted on average 495–610 ng/mL PEDF, which was less than healthy control hESC-RPEs (627–670 ng/mL). However, none of these differences was statistically significant as the highest significance in statistical analyses was *p* = 0.06 between type 2 diabetic and healthy control hiPSC-RPEs in NGM− (Figure 6). The type 2 diabetic and healthy control hiPSC-RPE cells secreted higher amounts of PEDF on the apical side than on the basal side (Appendix A). 

### 2.5. Effects of Glucose Concertation’s on Glucokinase Gene Expression 

Changes in the glucokinase (GCK) activity are involved in the glucose uptake, and its expression is affected by the insulin concentration [26]. As we did not detect any clear glucose concentration-dependent alterations in PEDF secretion, we ran a preliminary analysis (one biological replicate and three technical replicates) to see how glucose and insulin concentration effected *GCK* gene expression. The result showed that, except in the NG- condition, the expression of *GCK* was higher in type 2 diabetic cells than the healthy control in various conditions (Figure 7).

### 2.6. Effects of Glucose on Autophagy

The preliminary functionality assessment (with one biological replicate) of autophagic machinery in the hiPSC-RPE lines was done by evaluating autophagy marker protein LC3-II and p62/SQSTM1 expression from the Western blots of whole cell protein extracts (Figure 8a–c). Compared to control, autophagy stimuli 5-aminoimidazole-4-carboxyamide ribonucleoside (AICAR) increased the relative LC3-II level in both type 2 diabetic and healthy control hiPSC-RPEs in almost all glucose and insulin concentrations (Figure 8d–f) in healthy hiPSC-RPEs. Starvation, which is also known to activate autophagy, clearly decreased the relative amounts of LC3-II in type 2 diabetic and healthy control hiPSC-RPE and in all glucose and insulin concentrations (Figure 8d–f). We did not detect any difference in the LC3-II expression between diabetic or healthy control hiPSC-RPEs. Interestingly, in type 2 diabetic hiPSC-RPEs, there was clearly higher p62/SQSTM1 accumulation levels than in healthy control hiPSC-RPEs (Figure 8g–h). Additionally, the presence of insulin seemed to negatively affect the accumulation of p62/SQSTM in the cells, and particularly in type 2 diabetic hiPSC-RPEs.

### 2.7. Effects of Glucose and Insulin Concentration and Stress Inducers on MMP Secretion 

We carried out a preliminary functionality assessment, with one biological replicate, mimicking diabetic retinopathy conditions with the oxidative stress and cytokine stimuli. Thereafter, the activity of pro matrix metalloproteinases was evaluated. Both type 2 diabetic and healthy control hiPSC-RPEs expressed a band of 58.3 ± 5.3 kDa, corresponding to MMP2, and a band with molecular weight of 121.7 ± 8.0 kDa, corresponding to a high-molecular weight band in zymography (Figure 9a). Our analysis showed that the amount of pro-MMP2 was higher in healthy control hiPSC-RPEs than in type 2 diabetic hiPSC-RPEs (Figure 9b), while the pro-MMP9 was slightly more expressed in the diabetic than healthy control cells (Figure 9c). Tumor necrosis factor α (TNFα) treatment resulted in an increased pro-MMP2 expression in both type 2 diabetic and healthy hiPSC-RPEs; on the contrary, the oxidative stress with H_2_O_2_ resulted in decreased pro-MMP2 expression in both cell types (Figure 9d). The pro-MMP9 expression was increased in both diabetic and healthy hiPSC-RPEs by TNFα, and the highest increase was detected in HG+ in both cell types (Figure 9e). The H_2_O_2_ caused a modest decrease in pro-MMP9 expression in both cell types (Figure 9e). 

### 2.8. Effects of Glucose Concentration and Cytokine Stimulation on COL4A1, FN, and LAMA1 Gene Expression

We carried out preliminary functionality assessments (with one biological replicate and three technical replicates) mimicking diabetic retinopathy conditions, with the oxidative stress and cytokine stimuli. Thereafter, we analyzed the effects on the extracellular matrix protein expression known to be affected in diabetic retinopathy. Quantitative RT-PCR showed that the TNF-α treatment increased *COL4A1* gene expression by at least 1.5-fold compared to unstimulated cells in all other glucose and insulin concentrations except NG+ in type 2 diabetic hiPSC-RPEs (Figure 10a). In healthy control hiPSC-RPEs, TNF-α treatment resulted in a 2.5-fold increase in the *COL4A1* gene expression in NGM−, and 1.6-fold increase in NGM+, but a decrease in NG− grown cultures (Figure 10b). The expression of *FN1* was increased in diabetic hiPSC-RPEs after TNF-α treatment, and the highest induction was in NGM−, which had 4.8-fold increase, and found to be statistically significant (*p* = 0.05) (Figure 10c). The trend after the TNF-α treatment was the same in healthy control hiPSC-RPEs where the NGM− had the highest *FN1* induction (four-fold increase, *p* = 0.05) (Figure 10d). The *LAMA1* gene expression was modestly increased in type 2 diabetic hiPSC-RPEs after the TNF-α treatment (Figure 10e), whereas in healthy control hiPSC-RPEs, it resulted in to up to a five-fold increase (NGM− condition) gene expression, which was statistically significant (*p* = 0.05) (Figure 10f).

## 3. Discussion

High glucose concentrations in diabetic patients alter the functionality in retinal cells [1]. In the inner retina, the high glucose concentrations are shown to increase the permeability and decrease the viability of endothelial cells [1,27]. Although the effects of a high glucose concentration in the inner retina have been carefully assessed, there has been only few studies on its effects in the outer retina [1,2,3,6,7,8,10]. According to our knowledge, there are no previous publications in which the maturation of hiPSC-RPEs derived from type 2 diabetics and healthy controls have been assessed in high or normal glucose concentration in the presence or absence of added insulin. When RPE specific gene expression or PEDF secretion was assessed, both type 2 diabetic and healthy control hiPSC-RPEs matured well at all glucose concentrations in the presence or absence of added insulin. In previous studies in which human immortalized RPE cells have been cultured in different glucose concentrations, the glucose stimuli extended from three to 18 days [10,28,29,30,31]. Here the hiPSC-RPEs were cultivated in different glucose concentrations for up to five weeks, during which, the cells might have been adapted to different glucose concentrations. 

The tight junction integrity, also called the barrier function, is essential for the retinal functionality [3]. The barrier function of the epithelial cell layer can be analyzed by assessing TEER, which is a measure of electrical resistance, or cumulative permeability percentage, which is the flux of small molecular weight molecules. In previous studies, the retinas of diabetic rodents have been more permeable to solutes than healthy controls [9]. The high glucose concentration has also been shown to affect ARPE-19 cells, either by lowering TEER and increasing permeability of fluorescently labelled dextran [28,29], or by increasing the TEER and reducing the permeability [10]. In our study, the TEER increased over the five-week culture in all glucose concentrations in both type 2 diabetic and healthy control hiPSC-RPE cultures. After five weeks, the cumulative permeability percentage was close to zero, both in type 2 diabetic and healthy control hiPSC-RPEs, meaning that cultures, irrespective of the glucose or insulin concentrations, were highly polarized and non-permeable. Still, the type 2 diabetic hiPSC-RPEs in all assessed conditions were slightly more permeable than healthy control. 

Quite unexpectedly, after five weeks of culture, the epithelial layer was tighter in high glucose (HG+) concentration than in normal glucose (NGM−) concentration. This phenomenon was evident and statistically significant in both type 2 diabetic and healthy control hiPSC-RPEs (*p* = 0.003 or *p* = 0.009). It should be noted that when HG+ were compared to the NG−, the statistical significance dropped in both type 2 diabetic and healthy control hiPSC-RPEs from *p* ≈ 0.006 to *p* ≈ 0.4. This suggests that not only glucose concentration, but also osmolality, may have had an impact on hiPSC-RPE maturation. In addition, our data suggests that insulin concentration also had an effect on junctional integrity, as the NGM+ and NGM− had statistically significant differences, both in type 2 diabetic and healthy control hiPSC-RPE cultures.

PEDF secretion is a sign of a functional RPE [32]. The increase in PEDF secretion occurs concomitantly with the maturation of hiPSC-RPE cells [33]. Previously, it has been shown that high glucose concentration slightly increases the PEDF secretion in diabetic mice RPE [17] and in immortalized human RPE cells (D407) [16]. In our study, type 2 diabetic hiPSC-RPEs and healthy control hiPSC-RPEs secreted substantial amounts of PEDF (over 500 ng/mL), which can be taken as an indication of good maturation status. The diabetic hiPSC-RPEs secreted slightly lower amounts of PEDF than healthy control hiPSC-RPEs, but this difference was not statistically significant. We did not detect any decrease in the PEDF secretion of hiPSC-RPEs after cultivation in high glucose concentrations. 

Autophagy induction is an adaptive cellular response against cellular stress [34]. Previously, the autophagy was shown to be altered in immortalized RPE cells in a high glucose concentration [35,36]. Interestingly, in this study, with one biological replicate, we could not observe differences in autophagy in different glucose concentrations or between control and diabetic hiPSC-RPEs when LC3-II was used as a biomarker. However, a clear accumulation of p62/SQSTM1 was detected in diabetic hiPSC-RPEs compared to healthy control hiPSC-RPEs. In our hiPSC-RPEs, protein aggregates or damaged cell organelles were not detectable and the formation pattern of autophagosomes (LC3-II) was similar, which suggests another mechanism behind the elevated p62/SQSTM1 levels. In addition to autophagy regulation, the p62/SQSTM1 interacts with the antioxidative NRF-2/ARE pathway by disrupting the NRF-2-Keap1 complex [22,37]. This leads to nuclear localization of NRF-2 and induction of antioxidant response element-driven gene transcription, including p62/SQSTM1, and creating a positive feedback loop [21,37,38]. The upregulated p62/SQSTM1 levels in diabetic normal glucose hiPSC-RPEs might be due to an evoked energy depletion, together with induced antioxidant production via the NRF-2/ARE pathway.

The MMPs have long been thought to have a role in DR pathology [39], as increased MMP2 and MMP9 levels have been detected from the vitreous of DR patients [14] and also in the immortalized human RPE cells’ (ARPE-19) activity of gelatinase (MMP2 and MMP9), and expression of gelatinase (MMP2) [15] was shown to be increased after treatment with a high glucose concentration. In our pilot study in which we had one biological replicate, the type 2 diabetic hiPSC-RPEs expressed a lower amount of pro-MMP2, but a higher amount of pro-MMP9 compared to the healthy control hiPSC-RPEs in all studied glucose concentrations. This result suggests that for type 2 diabetic hiPSC-RPEs, the turnover of the extracellular matrix can be altered compared to a healthy control. 

TNFα is a multifunctional cytokine that has been implicated to have a role in pathologic events in early DR development [40]. Increased amounts of TNFα has been found in the vitreous of diabetic patients [41] and in diabetic rat retinas [42]. When the diabetic environment was simulated with the TNFα stimulation, we saw in the pilot assay with one biological replicate the increase in pro-MMP2 and pro-MMP9 expression in both diabetic and healthy control hiPSC-RPEs. The similar kind of induction in both type 2 diabetic and healthy control hiPSC-RPEs can be taken as an indication of the normal responsiveness of cell lines. 

The alterations in the expression of extracellular matrix components after the application of high glucose have been described previously in the literature. For example, in immortalized ARPE-19 cells, the high glucose induced the increased expression of *FN*, *COL4*, and *LAM* [12,13], but in RPE cells extracted from diabetic mice, there was no difference in the expression of *FN* or *COL4* under a high glucose condition [17]. In our study, in the pilot assay with one biological replicate and three technical replicates, we found no major differences between the type 2 diabetic and healthy control hiPSC-RPEs in *COL4A1*, *FN*, and *LAMA1* extracellular matrix protein expression under the different glucose concentrations. When we simulated the diabetic environment by adding TNFα to the cultures, which resulted in a statistically significant increase in *FN* gene expression in NGM− cultures in both diabetic and healthy control hiPSC-RPEs. This preliminary functionality assessment result carried out with one type 2 diabetic and one healthy control hiPSC-RPE line, which can be taken as an indication of normal responsiveness of these cells to cytokine stimuli.

## 4. Materials and Methods 

### 4.1. Human Induced Pluripotent Stem Cell Reprogramming

Five hiPSC lines were used in this study, three cell lines (UTA.08203.DMs UTA.08002.DMs, and UTA.10802.EURCCs) derived from type 2 diabetic patients and two lines (UTA.10212.EURCCs, UTA.10902.EURCCs) derived from control individuals. All the diabetic patients had nephropathy and neuropathy and universal atherosclerosis except one type 2 diabetic (UTA.10802.EURCCs). Diabetic retinopathy was diagnosed in two type 2 diabetic (UTA.08002.DMs, UTA.08203.DMs) patients. The fundus micrographs were from a type 2 diabetic (donor of UTA.08203.DMS), (see Figure 1). 

The hIPSC lines were derived from skin biopsy extracted fibroblasts and the pluripotency was induced by a Sendai-virus reprogramming kit (*OCT4*, *SOX2*, *KLF4*, *C-MYC*; CytoTuneTM; Life Technologies) as described elsewhere [43,44]. 

### 4.2. Human Induced Pluripotent Stem Cell Culture 

The hiPSC lines were cultured on mouse embryonic fibroblasts (MEFs, Applied StemCell Inc., Cat. No. ASF-1225) and maintained at +37 °C in 5% CO_2_ and were passaged on a weekly basis. Cells were cultured with Knock-Out Dulbecco’s Modified Eagle Medium (KO-DMEM, Gibco^®^) complemented with 20% Knock-Out Serum Replacement (Ko-SR, Gibco^®^), 2 mM Glutamax (Gibco^®^), 0.1 mM 2-mercaptoethanol (2-ME, Gibco^®^), 1% nonessential amino acids (NEAA, Lonza Group Ltd, Basel, Switzerland) and 50 U/mL penicillin/streptomycin (Lonza Group Ltd., Basel, Switzerland). Medium was supplemented with 4 ng/mL basic fibroblast growth factor (bFGF, R & D Systems Inc., Minneapolis, MN, USA). Cell lines were regularly checked for mycoplasma contamination. 

### 4.3. Human Induced Pluripotent Stem Cell Characterization

All the hiPSC lines were characterized in detail as we described before [45]. Briefly, the absence of imported exogenes (*OCT4*, *SOX2*, *KLF4*, and *c-MYC*) and the expression of endogenous pluripotency genes (*OCT3/4*, *NANOG*, *SOX2*, *REX1*, and *c-MYC*) was evaluated using PCR. The protein expression of pluripotency markers OCT-3/4, Nanog, SSEA-4, SOX2, TRA 1-60, and TRA 1-81, was confirmed using indirect immunofluorescence staining. In addition, the pluripotency of the hiPSCs was verified in vitro via the formation of embryoid bodies (EBs). Then, the expression of marker genes characteristic of the endoderm (*SOX17* or *AFP*), mesoderm (*KDR* or *ACTC1*), or ectoderm (*SOX1*, *PAX6*, *Nestin*, or *Musashi*) were studied from the extracted RNAs of the EBs and *GAPDH* was used as an endogenous control. The list of primer sequences for pluripotency genes, marker genes of the three germ layers, and the list of primary and secondary antibodies has been published before [46]. Normal karyotype of the hiPSC lines was evaluated in the Finnish Microarray and Sequencing Centre by performing genome-wide screening for gross chromosomal abnormalities with KaryoLite BoBs (Perkin Elmer, 4501–0010) as described elsewhere [47].

### 4.4. Differentiation of Human Induced Pluripotent Stem Cells to Retinal Pigment Epithelial Cells

The differentiation towards RPE started by dissociating the undifferentiated cells cultured on top of mitotically inactivated mitomycin (10 mg/mL, Sigma-Aldrich, St. Louis, MO, USA) treated human foreskin fibroblasts feeder cell (CRL-2429TM, ATCC, Manassas, VA, USA) hiPSC colonies that were enzymatically with TrypLE™ Select Enzyme (Gibco^®^, Thermo Fisher Scientific). These were transferred to Corning^®^ Costar^®^ Ultra-Low attachment plates, and grown in Knock-Out™ Dulbecco’s Modified Eagle’s Medium (DMEM) supplemented with 20% Knock-Out™ SR (KO-SR), 1% MEM non-essential amino acids, 0.1 mM 2-mercaptoethanol, 2 mM GlutaMAX™, and 50 U/mL penicillin-streptomycin (all from Gibco^®^, Thermo Fisher Scientific, Grand Island, NY, USA). The embryoid body (EB) formation was enhanced using overnight treatment with 5 μM blebbistatin (Sigma-Aldrich, St. Louis, MO, USA) at +37 °C. Thereafter, the spontaneous RPE differentiation was induced by reducing the KO-SR to 15%. The EBs differentiation was for 56 to 70 days. During this period, medium was replenished thrice a week. After this period, pigmented areas were manually separated from EBs with a scalpel, dissociated with TrypLE™ Select Enzyme, and acquired single-cell suspension filtered through a 100 μm BD Falcon cell strainer (BD Biosciences, San Jose, CA, USA), and replated onto well plates coated with human Collagen IV(COLIV, 5 μg/cm^2^, Sigma-Aldrich) to expand cell numbers and purify the cell population. To expand and purify the culture further, this replating was repeated after 33 to 70 days.

### 4.5. Cultivation of hiPSC-RPEs for the Experiments

After the expansion of cells, the hiPSC-RPEs were dissociated with Trypsin–ethylenediaminetetraacetic acid (EDTA), filtered through a strainer and counted to be plated for the experiments on polyethylene terephthalate (PET) hanging cell culture inserts with a 1.0 μm pore size (Merck Millipore Corporate, Billerica, MA, USA) which were coated on 10 μg/cm^2^ COL IV (Sigma-Aldrich St. Louis, MO, USA) and 0.75 μg/cm^2^ Laminin 521 (LN-521; Biolamina, Sundbyberg, Sweden). Plated cells were grown in No-glucose DMEM medium with 2 mM GlutaMAX™, and 50 U/mL penicillin-streptomycin (all from Gibco, Thermo Fisher Scientific, Grand Island, NY, USA) and 10% heat-inactivated FBS. This base was supplemented with d-glucose or d-mannitol (both from Sigma-Aldrich) or +/− added human insulin (Gibco^®^, Thermo Fisher Scientific). The high glucose (HG) medium contained 25 mM of glucose, normal glucose (NG) medium contained 5 mM of glucose, isotonically balanced normal glucose medium (NGM) contained 5 mM of glucose and 19.5 mM mannitol. To mimic the amount of insulin after the meal/fasting to the HG medium, the amount of added insulin was higher (IU/mL equivalent to 6.94 ng/mL), whereas in the NG or NGM medium, the amount of added insulin was lower (25 IU/mL, equivalent to 0.87 ng/mL). Immediately after plating on PET inserts, the cells were grown in six different media with different glucose concentrations and with or without added insulin (HG−, HG+, NGM−, NGM+, NG−, NG+) for 41 +/− 5 days, and the median was 35 days. The entire culture period from the EB culture until the end of experiment was from 161 to 286 days, median was 188 days.

### 4.6. Experimental Treatments

The cellular stress was induced using 24 h treatment with 10 ng tumor necrosis factor α (TNFα, Peprotech, London, U.K.), and a 1 h treatment with 300 mM hydrogen peroxide (H_2_O_2_, Sigma-Aldrich) followed with a 23-h chase. The treatments for the autophagy function assessments were 24-h treatments were starvation, which was done by omitting the FBS from the culture medium [48] or by adding 20 μM Resveratrol (Sigma-Aldrich) or 2 mM AICAR (5-aminoimidazole-4-carboxyamide ribonucleoside, Toronto Research Chemical, North York, ON, Canada). 

### 4.7. Indirect Immunofluorescence Staining

The same cultures that were subjected to the permeability test were used for indirect immunofluorescent staining. Samples were first washed in 1× PBS, fixed with 4% paraformaldehyde for 10 min at room temperature (RT), and followed with four 1× PBS washes and permeabilized with 0.1% TritonX-100 in 1× PBS for 10 min at RT. After that, the samples were washed repeatedly with PBS, and 3% bovine serum albumin (BSA) in PBS was added to the samples for 1–1.5 h at RT or overnight at 4 °C to block the nonspecific binding sites. The samples were then incubated with primary antibody against tight junction specific with mouse-anti-ZO-1 (1:250 Invitrogen, Carlsbad, CA, USA) in 0.5% BSA-PBS for 1 h at RT, and thereafter followed by four washes with 1× PBS. Samples were then incubated with donkey-anti-mouse (Life Technologies, Paisley, UK) secondary antibody at a dilution of 1:1000 in 0.5% BSA-PBS for 1 hour at RT. Finally, the samples were washed four times with 1× PBS mounted between two cover glasses with Vectashield^®^ mounting medium with 4′,5-diamidino-2-phenylindole (DAPI) (Vector Laboratories Inc., Burlingame, CA, USA). The visualization and imaging of the stained samples was carried out with an AxioScope A1 (Carl Zeiss, Jena, Germany) using a magnification of 40× and resolution of 1200 × 1200.

### 4.8. Preparation of Transmission Electron Microscopy Samples

After 35 days of culture in different glucose concentrations, cells were fixed for 2 h at RT with 2% glutaraldehyde (Electron Microscopy Sciences, Hatfield, PA, USA) in a 0.1 M phosphate buffer, and washed thereafter five times with 0.1 M phosphate buffer. The samples were postfixed with 1% osmium tetroxide (Ladd Research, Williston, VT, USA) for 2 h at RT and thereafter washed thoroughly with deionised water. The samples were then dehydrated through acetone series: 3 × 10 min 70% acetone, 3 × 10 min 94% acetone, and 1 × 20 min and 1 × 30 min absolute acetone (J.T. Baker; Avantor Performance Materials, B.V. Deventer, Arnhem, The Netherlands). Samples were impregnated with a 1:1 mixture of absolute acetone and epoxy resin (Ladd Research, Williston, VT, USA) for 1.5 h at RT. Thereafter, the excess acetone-epoxy resin solution was removed and replaced with pure epoxy resin solution. Embedding with pure epoxy resin was done overnight at RT, and polymerization took place for 48 h at 60 °C. Thin sections were stained with 1% uranyl acetate for 30 min and with 0.4% lead citrate (Fluk, Steinheim, Switzerland) for 5 min. Samples were examined and imaged with a JEM-2100F TEM (Jeol Ltd., Tokyo, Japan).

### 4.9. RNA Extraction and cDNA Synthesis

For the gene expression analyses, the total RNA was extracted from cell samples with a NucleoSpin XS-kit (Macherey-Nagel, GmbH & Co., Düren, Germany) according to the manufacturer’s instructions. The RNA concentration and its quality were assessed using a NanoDrop 1000 spectrophotometer (NanoDrop Technologies, Wilmington, DE, USA). RNA (40 ng) was reverse-transcribed to complementary DNA using MultiScribe Reverse Transcriptase (Applied Biosystems, Foster City, CA, USA) according to the manufacturer’s instructions in the presence of an RNase inhibitor. 

### 4.10. PCR Reaction

The RPE characteristics of hiPSC-RPEs was analyzed with RT-PCR using complementary DNA as a template. The reaction was done using 5 μM primers specific for specific genes (Biomers.net GmbH, Söflinger, Germany, Table 1), 5 U/μL Taq DNA Polymerase (Fermentas, Thermo Fisher Scientific Inc., Leicestershire, U.K.) in PCR MasterCycler ep gradient (Eppendorf AG, Hamburg, Germany) according the protocol: 95 °C 3 min, 95 °C 30 s, annealing 30 s, 72 °C 1 min, 72 °C 5 min, for 38 cycles. Annealing temperatures and primer sequences are presented in Table 1. PCR products were resolved in 2% agarose gels with a 50-bp DNA ladder (MassRulerTM DNA Ladder Mix, Fermentas, Thermo Fisher Scientific Inc., Leicestershire, U.K.). The products were visualized with the Quantity One 4.5.2. Basic program (Bio-Rad Laboratories, Inc., Hercules, CA, USA).

### 4.11. Quantitative RT-PCR

The effects of different glucose concentrations was assessed by analyzing the expression of *glucokinase* (*GCK*, Hs01564555_m1) which was done similarly as previously described in Kiamehr et al. [49]. The effects of different glucose concentrations or glucose concentrations together with cellular stress using TNFα and H_2_O_2_ was analyzed with qRT-PCR with TaqMan^®^ gene expression assays (Applied Biosystems, Inc., Foster City, CA, USA) using FAM labels. The expression of collagen 4A1 (*COL4A1*, Hs00266237_m1), *fibronectin 1* (*FN1*, Hs00365052_m1), and *laminin A1* (*LAMA1*, Hs00300550_m1) genes were analyzed and compared against the *glyceraldehyde 3-phosphate dehydrogenase* (*GAPDH*; Hs99999905_m1), which was used as an endogenous control. Samples and template-less controls of *COL1A4*, *FN1*, and *LAMA1* were run in triplicate using the 7300 Real-Time PCR system (Applied Biosystems, Inc. Inc., Foster City, CA, USA) with the following program: 2 min at 50 °C, 10 min at 95 °C; 40 cycles of 15 s at 95 °C, and finally 1 min at 60 °C. Results of *COL1A4*, *FN1*, and *LAMA1* were analyzed using 7300 System SDS Software 2.4 (Applied Biosystems, Inc. Inc., Foster City, CA, USA). The relative quantification of each gene was calculated using *C*_t_ values and the 2^−ΔΔ*C*t^ method [50] using *GAPDH* as a calibrator. 

### 4.12. Trans-Epithelial Electrical Resistance

The barrier properties of the samples cultured in different glucose concentrations were evaluated using trans-epithelial electrical resistance (TEER) analysis once a week during the four-week period after plating hiPSC-RPEs on the PET insert. TEER values (Ω∙cm^2^) were calculated by multiplying the result by the surface area of the insert. TEER values were obtained from three to five individual experiments with all hiPSC-RPE lines with two to three parallel samples, and two technical replicates.

### 4.13. Permeability Tests

After the five-week culture period, the cell culture inserts were cut from the holder and clamped to a P2307 slider (Physiologic Instruments, San Diego, CA, USA) and placed into the Ussing Chamber device (EM-CSYS-8, Physiologic Instruments) with P2300 EasyMount Diffusion Chambers with the aperture of 0.03 cm^2^ to measure the barrier properties. Assessments were done similarly as in Skottman et al. [51], except that the assessments were carried out in culture media containing the same concentration of glucose in which the sample was differentiated. Briefly, 1 mg/mL of 4 kDa fluorescein isothiocyanate–dextran (FD4, Sigma-Aldrich) was used as the test molecule, and placed on the donor side. Blank samples were taken prior to the test from both donor and acceptor media. The pH in the chambers was kept constant with carbon dioxide gas (5% CO_2_, 10% O_2_, 85% N_2_). Once an hour for 4 hours, two parallel 100 μL samples were taken from the acceptor side of the chamber and 200 μL of fresh medium on was added to balance the removed volume. The spectrophotometric assessment was done with Wallac Victor2 1420 multilabel counter spectrophotometer (Turku, Finland). The cumulative permeability value was calculated from the average value of the two parallel spectrophotometer measurement values from which the value of pure medium (i.e., blank) was subtracted. This value was then divided by the total volume of the acceptor side. The dilution effect of the added fresh medium after every hour was taken into account in this step. Finally, this volume-adjusted value was divided by the spectrophotometer value of the exposure medium and multiplied by 100%.

### 4.14. Enzyme-Linked Immunosorbent Assay

After the five-week culture period of hiPSC-RPEs on the 0.3 cm^2^ PET insert in different glucose concentrations, the concentration of secreted PEDF was determined 24 h after the medium change with a PEDF Enzyme-Linked Immunosorbent Assay (ELISA, BioVendor, Brno, Czech Republic) according to the manufacturer’s instructions.

### 4.15. Western Blotting

For the Western blotting, the five-week cultured cell samples were washed once with PBS (Lonza Group Ltd., Walkersville, MD, USA) and lysed in M-PER lysis buffer (Thermo Scientific, Waltham, MA, USA) according to the manufacturer’s instructions. A total of 25–30 μg of whole cell extracts were run in 15 % sodium dodecyl sulphate PAGE (SDS–PAGE) gels, wet-blotted to nitrocellulose membranes (Amersham, Pittsburgh, PA, USA), and the unspecific binding sites on membranes blocked using 3% skimmed milk powder in 0.3 % Tween-20/PBS at RT for 1.5 h. The labelling times and primary antibodies against autophagy specific proteins p62/SQSTM1 and LC3, as well as housekeeping protein α-tubulin, are presented in Table 2. After primary antibody incubations, the membranes were washed three times for 5 min with the same buffer the antibodies were diluted with. The secondary antibodies and dilution factors are presented in Table 2. After staining, similar to the previous washes, the membranes were washed three times for 5 min with the same buffer the antibodies were diluted with. Protein–antibody complexes were detected with an enhanced chemiluminescent assay for horseradish peroxidase (Millipore, Billerica, MA, USA).

### 4.16. Zymography

Medium from the apical and basal compartments was collected 24 hours after the medium change. Zymography was conducted as previously described [52]. In brief, the medium from the apical and basal compartments was collected and 10 L conditioned media was mixed with 20 L nonreducing zymogram sample buffer and loaded into the 0.1% gelatin-containing SDS-PAGE gel (10% Ready Gel; BioRad, München, Germany). After electrophoresis, gels were incubated with renaturation and development buffer (both BioRad). Afterward, gels were stained with Coomassie blue (BioRad) and destained with a destaining solution (BioRad). Digital images of the gels were obtained with a Chemibis chemoluminescence system (Biostep, Jena, Germany). For quantification, the inverted gelatinase bands were analyzed with one-dimensional gel analysis software (TotalLab TL100; TotalLab Ltd., Newcastle, U.K.). The molecular weight and band volume (density) was assessed. When the effects of different glucose and insulin concentrations were compared, the 100 kDa molecular marker (MW) was used as the calibrator. When the effects of cytokine or oxidative stress were evaluated, the b band of the untreated control of the respective MMP was set as 1 (arbitrary unit). 

### 4.17. Statistical Analyses

The statistical significance of numerical data were analyzed using PASW Statistics, version 18, with a two tailed Mann–Whitney U test. The number of replicates is indicated in the figure legends. 

### 4.18. Ethical Issues

The study was approved by the ethical committee of Pirkanmaa Hospital District (R12123) and written consent was obtained from all fibroblast donors. All the patients were over 18 years old. 

## 5. Conclusions

The type 2 diabetic hiPSC-RPE cells exhibited RPE type gene expression and PEDF secretion. Diabetic hiPSC-RPEs had a higher cumulative permeability than normal controls. Added insulin increased the epithelial layer tightness in normal glucose concentrations, and the effect was clearer in type 2 diabetic hiPSC-RPEs than in healthy control hiPSC-RPEs. The preliminary functionality assessment under oxidative stress and autophagy and cytokine stimuli showed that the autophagic stimulation had no effect on LC3-II but induced accumulation of p62/SQSTM1, and cytokine stimuli decreased pro-MMP2 expression and increased pro-MMP9 expression in type 2 diabetic hiPSC-RPEs. These results suggest that the used cell model has potential to study diabetes-derived cellular stress alterations. 

## Figures and Tables

**Figure 1 ijms-20-03773-f001:**
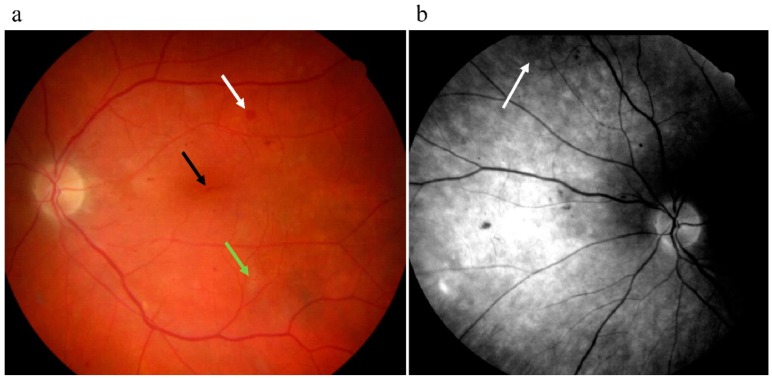
(**a**) Fundus photograph of donor of UTA.08203.EURCCS cell line having severe diabetes retinopathy with a microaneurysm (black arrow), hemorrhage (white arrow), and soft exudate (green arrow). (**b**) A red free fundus photograph from the same patient with intraretinal microvascular abnormalities (IRMA) pointed with an arrow.

**Figure 2 ijms-20-03773-f002:**
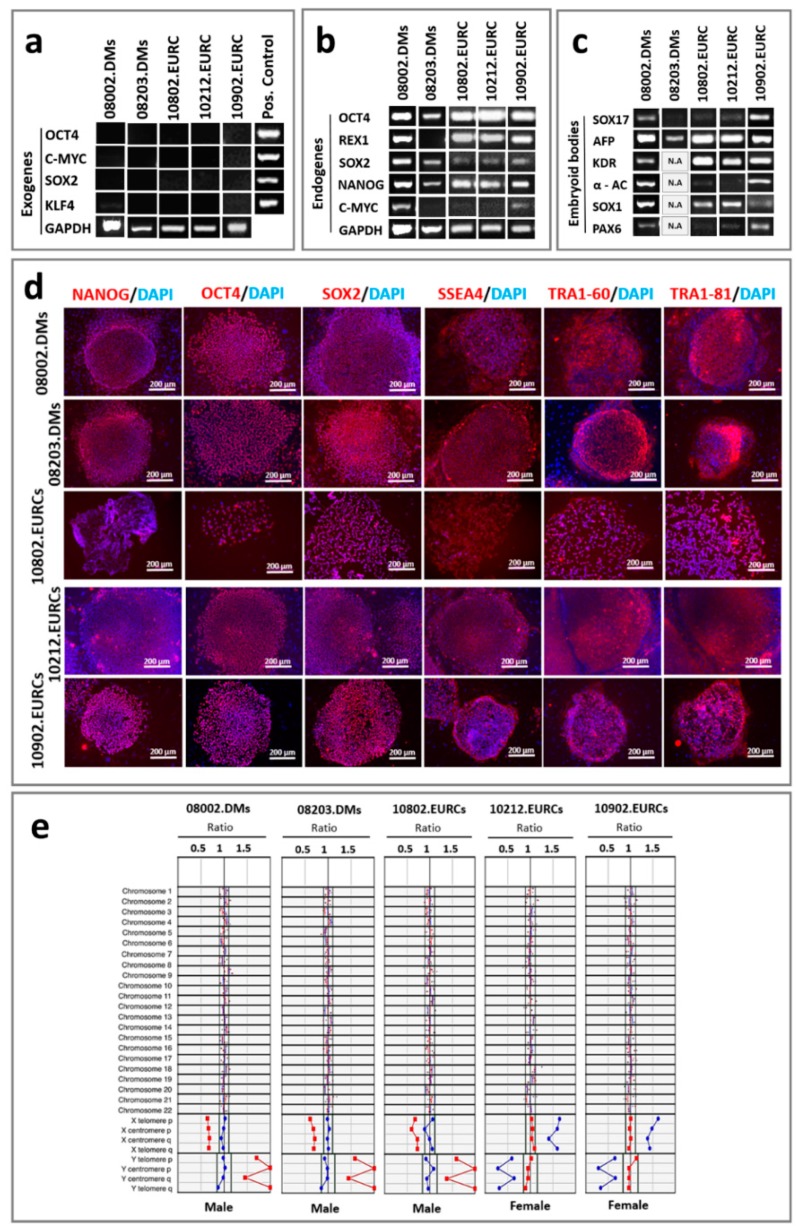
Characterization of the human induced pluripotent stem cell (hiPSC) line gene and protein expression and analyses of their karyotype. (**a**) The virally transferred Sendai exogenes *OCT4*, *c-MYC*, *SOX2*, and *KLF4* were silenced in all the iPSC lines. The RNA that was extracted one week after the viral transduction was taken as a positive control. *GAPDH* was used as a housekeeping gene. (**b**) All the five hiPSC lines expressed endogenous pluripotency genes *OCT4*, *REX1*, *SOX2*, *NANOG*, and *c-MYC*. (**c**) All five hiPSC lines formed embryoid bodies (EBs) expressing at least one marker from each of the germ layers: endoderm (AFP, SOX17), mesoderm (KDR, α-cardiac actin α AC), and ectoderm (SOX1, POX6). (**d**) The expression of pluripotency markers in all iPSC lines were confirmed at the protein level using indirect immunofluorescence staining. The red color indicates the specific signal for NANOG, OCT4, SOX2, SSEA4, TRA1-60, and TRA1-81, and the blue color indicates the nuclei stained using DAPI (4′,6-diamidino-2-phenylidole). The scale bar represents 200 µm. (**e**) Karyotype analyses of five hiPSC lines. Red and blue dots indicate chromosomal signal ratios of sample DNA against male (blue) or female (red) reference normal karyotype DNA, as detected using a KaryoLiteTM BoBs™ assay. Signal from normal chromosomes should lie inside the reference area around value 1, while with an abnormal karyotype, signals lie outside the reference area. Cell lines UTA.08002.DMs, UTA.08203.DMs, and UTA.10802.EURCCs showed a normal male karyotype, and cell lines UTA.10212.EURCCs, and UTA.10902.EURCCs showed a normal female karyotype.

**Figure 3 ijms-20-03773-f003:**
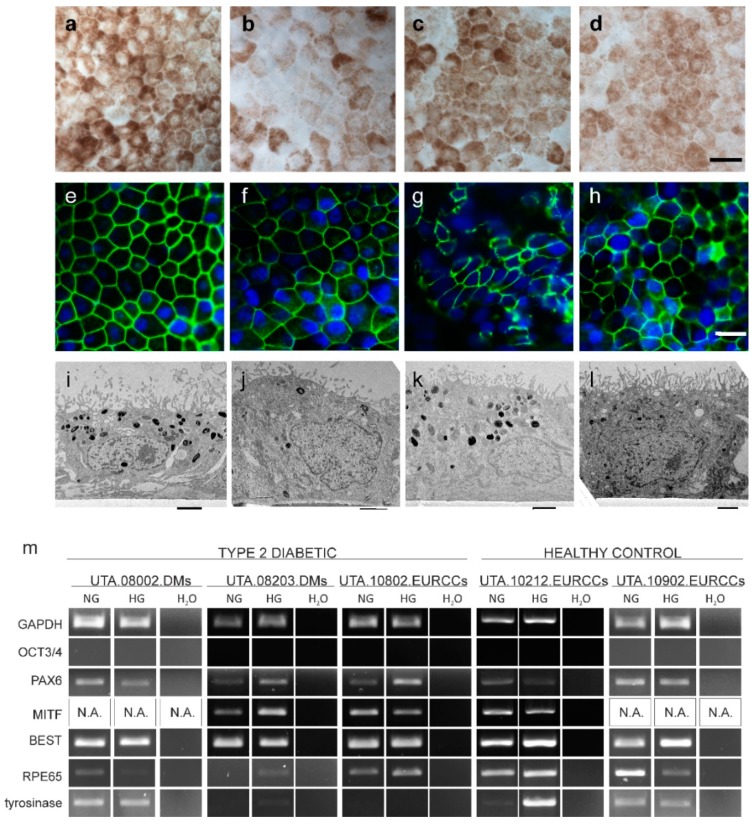
Characterization of RPE specific features. (**a**–**d**) The phase contrast micrographs taken after the five weeks of cultivation showed a substantial amount of pigmentation in type 2 diabetic and healthy control cell lines (**a**) type 2 diabetic, UTA.08002.DMs; (**b**) type 2 diabetic, UTA.08203.DMs; (**c**) type 2 diabetic, UTA.10802.EURCCS; and (**d**) healthy control, UTA.10902.EURCCs. The scale bar represents 20 μm. (**e**–**h**) Immunofluorescent staining of tight junction ZO-1 localized to the cell edges. (**e**) type 2 diabetic, UTA.08002.DMs; (**f**) type 2 diabetic, UTA.08203.DMs; (**g**) type 2 diabetic, UTA.10802.EURCCS; and (h) healthy control, UTA.10902.EURCCs. The scale bar represents 20 μm. Transmission electron micrographs reveal that hiPSC-RPEs were growing in a monolayer, have melanin granules and a thick brush border. (**i**) type 2 diabetic, UTA.08002.DMs; (**j**) type 2 diabetic, UTA.08203.DMs; (**k**) type 2 diabetic, UTA.10802.EURCCS; and (**l**) healthy control, UTA.10902.EURCCs. The scale bar represents 2 μm. (**m**) The RPE cells differentiated from type 2 diabetic (UTA.08002.DMs, UTA.08203.DMs, UTA.10802.EURCCs) and healthy control (UTA.10212.EURCCs, UTA.10902.EURCCs) cells during five weeks of time in normal (NG) or high (HG) glucose concentration, as studied in RT-PCR. The hiPSCs expressed the housekeeping gene *GAPDH*, and eye-specific lineage marker *PAX6*, as well as RPE-specific marker *Bestrophin* (*BEST*), and low amounts of *tyrosinase*, but no pluripotency marker *OCT3/4*. The negative controls (marked with H_2_O) are shown on the right-hand side.

**Figure 4 ijms-20-03773-f004:**
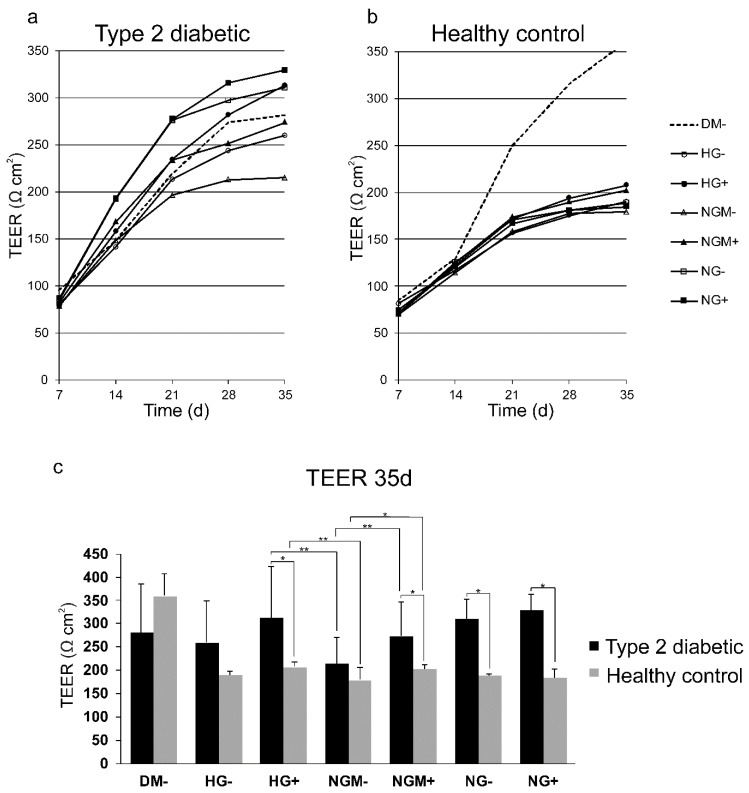
Development of barrier function in diabetic and healthy control hiPSC-RPEs. Cells were cultured for 5 weeks in different glucose and insulin concentrations (see treatments and abbreviations below). The development of trans-epithelial electrical resistance (TEER) during the five-weeks culture: (**a**) represents three type 2 diabetic cell lines (UTA.08002.DMs, UTA.08203.DMs, UTA.10802.EURCCs) (*n* = 3–4 biological, and 2 technical replicates); (**b**) represents one healthy control cell line (UTA.10902.EURCCs) (*n* = 3 biological, and 2 technical replicates). (**c**) The TEER after 35 days of culture. Data are presented as mean ± SD. Statistical significance * *p* < 0.05, ** *p* < 0.01. HG represents high glucose (25 mM); NG represents normal glucose (5 mM); NGM represents normal glucose (5 mM) balanced with mannitol (19.5 mM). DM− is the control culture medium, which is ordinarily used for hiPSC maturation.

**Figure 5 ijms-20-03773-f005:**
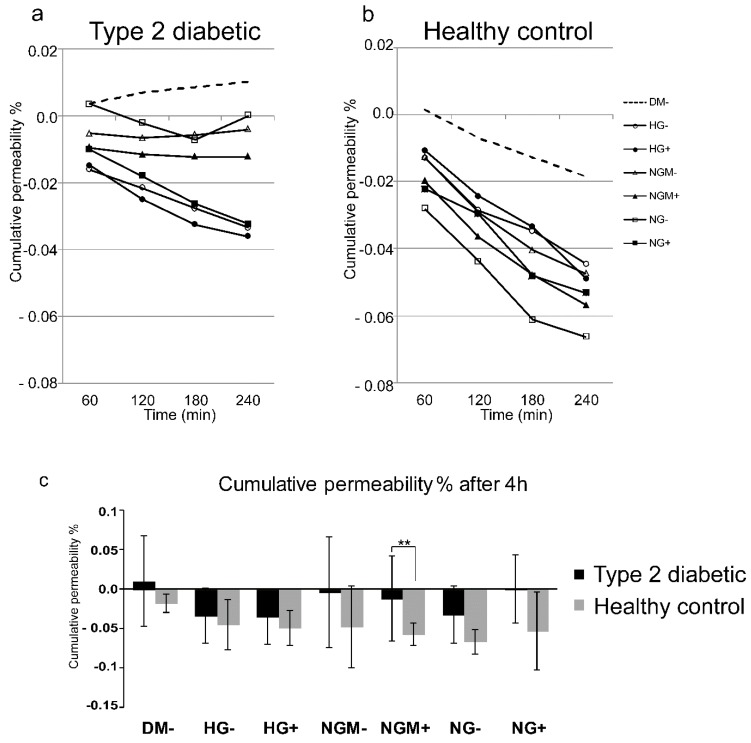
Cumulative permeability in diabetic and healthy control hiPSC-RPEs. After the 5-week culture, the hiPSC-RPEs were subjected to permeability analysis (**a**–**c**). The cumulative transport percentage of 4 kDa Fitc dextran (FD4) from the apical to basal side of hiPSC-RPE in an Ussing chamber system over 240 min in (**a**) type 2 diabetic, and (**b**) healthy control cells. The endpoint measurement of cumulative transport percentage at the 240 min time point is shown in separate graphs presented in (**c**). Data are presented as mean ± SD. Statistical significance *p* < 0.01 marked as **. HG represents high glucose (25 mM); NG represents normal glucose (5 mM); NGM represents normal glucose (5 mM) balanced with mannitol (19.5 mM). All conditions had +/− added insulin. DM− is the control culture medium, which is ordinarily used for hiPSC maturation.

**Figure 6 ijms-20-03773-f006:**
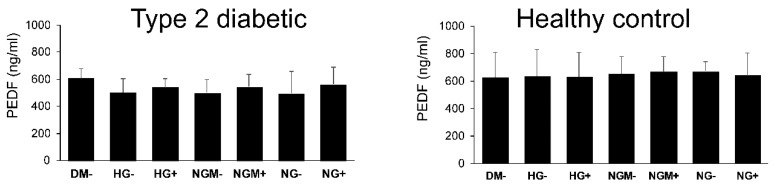
The effects of glucose and insulin concentration and extracellular stress on pigment epithelial growth factor (PEDF) secretion. The amount of PEDF was assessed using enzyme-linked immunosorbent assay (ELISA) from medium samples after a 24 h collection of apical and basal side (1 + 1). Medium was from hiPSC-RPEs derived from type 2 diabetic (UTA.08002.DMs, UTA.08203.DMs, and UTA.10802.EURCCs) and healthy control (UTA.10902.EURCCs) patients (1–3 biological replicates, 1 technical replicate).

**Figure 7 ijms-20-03773-f007:**
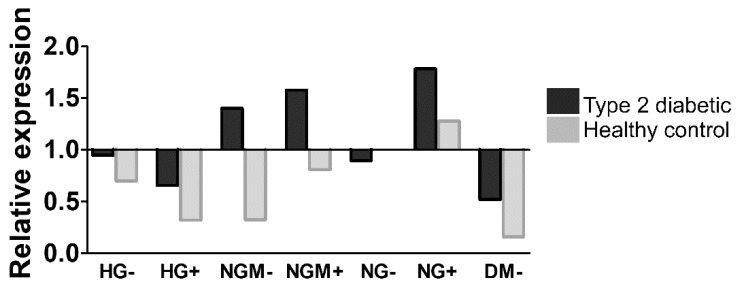
The preliminary assessment of *glucokinase* (*GCK*) gene expression in different glucose and insulin concentrations. Expression of *GCK* in type 2 diabetic (UTA.08002.DMs) or healthy control (UTA.10902.EURCCs) patient-derived hiPSC-RPEs (three technical replicates, one biological replicate) was studied when cultured in different glucose and insulin concentrations (see treatments and abbreviations below). Treatments and their abbreviations: High glucose (HG, 25mM), normal glucose 5 mM balanced with mannitol 19.5 mM (NGM), or normal glucose (NG, 5 mM) in the presence or absence (+/−) of added insulin. Each bar represents the medium of three technical replicates.

**Figure 8 ijms-20-03773-f008:**
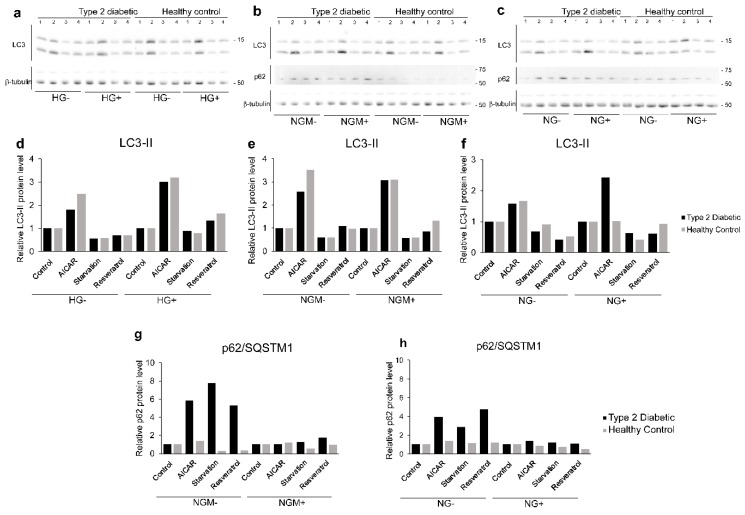
Activation of autophagy was analyzed using hiPSC-RPE cells derived from type 2 diabetic (UTA.08203.DMs) or healthy control (UTA.10902.EURCCs) patients cultured in high glucose (HG, 25 mM, (**a**,**d**)), normal glucose (5 mM) balanced with mannitol (19.5 mM) (NGM, **b**,**e**,**g**), or normal glucose (NG, 5 mM, (**c**,**f**,**h**)) for five weeks. Thereafter, cultures were subjected to 5-aminoimidazole-4-carboxyamide ribonucleoside (AICAR, 2 mM), starvation (no serum), or resveratrol (20 mM) for 24 h, and whole proteins were extracted and analyzed using Western blots labelled with LC3, p62/SQSTM1, and α-tubulin. The Western blots are represented in (**a**–**c**). The line on the left denotes the size of the blot. The α-tubulin band was used as a loading control. The values are correlated to the untreated sample, which is marked as 1. AICAR increased the relative LC3-II level in both diabetic and healthy control hiPSC-RPEs and in all glucose and insulin concentrations (**d**–**f**). Starvation resulted in decreased relative amounts of LC3-II in both diabetic and healthy control hiPSC-RPE in all glucose and insulin concentrations (**d**–**f**). In diabetic hiPSC-RPEs, the levels of relative p62/SQSTM1 were clearly higher than for healthy control cells (**g**–**h**).

**Figure 9 ijms-20-03773-f009:**
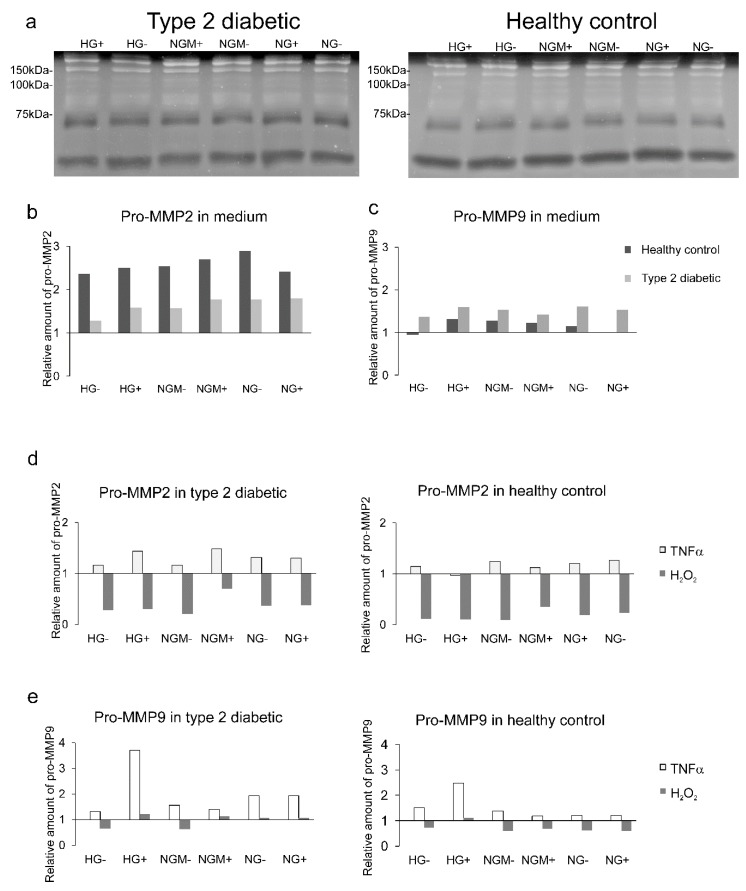
Effects of different glucose concentrations, and cytokine and oxidative stress on pro-MMP secretion. The type 2 diabetic (UTA.08203.DMs) and healthy control (UTA.10212.EURCCs and UTA.10902.EURCCs) hiPSC-RPEs were cultured in different glucose and insulin concentrations (see treatments and abbreviations below) in the presence or absence (+/−) of added insulin (**a**–**d**), and in the presence or absence of TNFα (48 h) or H_2_O_2_ (1 h + 23 h). (**a**) represents the exemplary zymographies with diabetic (left) and healthy (right) control samples. The densitometric evaluation of zymograms are shown in panels (**b**–**d**). The densitometric analysis of pro-MMP2 (**b**) and pro-MMP9 were normalized to a 100 kDa marker (**c**). The densitometric evaluation of pro-MMP2 after 10 ng/mL TNFα (48h) or 300 mM H_2_O_2_ (1 h + 23 h) in diabetic or healthy control hiPSC-RPE, and the values are correlated to the untreated sample, which is marked as 1 (**d**). The densitometric evaluation of pro-High molecular weight matrix metalloproteinase (HMW MMP) after 10 ng/mL TNFα or 300 mM H_2_O_2_ (1 h + 23 h) in diabetic or healthy control hiPSC-RPEs, where the values are correlated to the untreated sample, which is marked as 1 (**e**). Treatments and their abbreviations: high glucose (HG, 25 mM), normal glucose 5 mM balanced with mannitol 19.5 mM (NGM), or normal glucose (NG, 5 mM) in the presence or absence (+/−) of added insulin.

**Figure 10 ijms-20-03773-f010:**
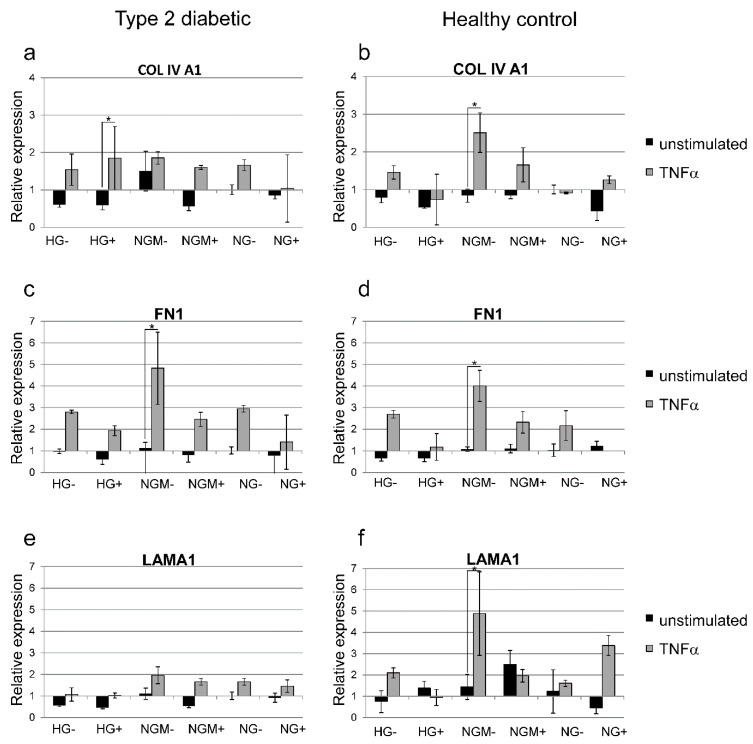
Effects of different glucose concentrations, and cytokine and oxidative stress on *Collagen IV* (*COL4A1*), *fibronectin* (*FN1*), and *laminin A1* (*LAMA1*) gene expression in hiPSC-RPEs derived from type 2 diabetic (UTA.08203.DMs: (**a**,**c**,**e**)) or healthy control (UTA.10902.EURCCs: (**b**,**d**,**f**)) patients. The hiPSC-RPEs were cultured in different glucose and insulin concentrations (see treatments and abbreviations below) in the presence or absence (+/−) of added insulin, and in the presence or absence of TNFα (48 h treatment). Overall, the 48 h TNFα treatment induced *COLA1*, *FN1*, and *LAMA1* expression. The housekeeping gene *GAPDH* was used as a loading control and calibrator. Data are expressed as mean ± SD of three replicates. The statistical significance is presented against the control: * *p* ≤ 0.05. Treatments and their abbreviations: high glucose (HG, 25 mM), normal glucose 5 mM balanced with mannitol 19.5 mM (NGM), or normal glucose (NG, 5 mM) in the presence or absence (+/−) of added insulin.

**Table 1 ijms-20-03773-t001:** Reverse-transcriptase–PCR primer sequences, product lengths (bp) and used annealing temperatures (Tm). Primer sequences (5′ > 3′).

Gene	Forward	Reverse	bp	Tm
*GAPDH*	GTTCGACAGTCAGCCGCATC	GGAATTTGCCATGGGTGGA	229	55
*OCT 3/4*	CGTGAAGCTGGAGAAGGAGAAGCTG	AAGGGCCGCAGCTTACACATGTTC	245	55
*PAX6*	AACAGACACAGCCCTCACAAACA	CGGGAACTTGAACTGGAACTGAC	274	60
*BEST*	GAATTTGCAGGTGTCCCTGT	ATCAGGAGGACGAGGAGGAT	214	60
*RPE65*	TCC CCA ATA CAA CTG CCA CT	CAC CACC ACA CTC AGA ACT A	316	52
*Tyrosinase*	TGC CAA CGA TCC TAT CTT CC	GAC ACA GCA AGC TCA CAA GC	316	52

**Table 2 ijms-20-03773-t002:** The antibodies and dilution buffers used for Western blotting.

Protein	Primary Antibody Staining		Secondary Antibody Staining		Buffer Used for Dilution and Washing
	Antibody	Cat, Producer, Dilution	Antibody	Producer, Dilution, Time	
p62	Mouse monoclonal p62 antibody	sc-28359, Santa Cruz Biotechnology Inc, CA, USA, 1:1000	Horseradish peroxidase-conjugated anti-mouse IgG	GE Healthcare, Little Chalfont, Buckinghamshire, UK, 1:10,000, 2 h	0.5% BSA in 0.3% Tween-20/PBS
LC3	Rabbit polyclonal LC3 antibody	3868, Cell Signaling, Danvers, MA, USA, 1:1000	Horseradish peroxidase-conjugated anti-rabbit IgG	Novex™, ThermoFisher Scientific, 1:10,000, 2 h	5% BSA in 0.1% Tween-20/TBS (tris-buffered saline) overnight at 4 °C
α-tubulin	Mouse monoclonal alpha-tubulin antibody	T5168, Sigma-Aldrich, 1:8000	Horseradish peroxidase-conjugated anti-rabbit IgG	GE Healthcare, Little Chalfont, Buckinghamshire, UK, 1:10,000, 1 h	1% milk powder in 0.05 % Tween-20/PBS for 1hrou at RT

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
