# Peer review of "Compromised Barrier Function in Human Induced Pluripotent Stem-Cell-Derived Retinal Pigment Epithelial Cells from Type 2 Diabetic Patients"

_ijms, 2019, doi:10.3390/ijms20153773_

Round 1

Reviewer 1 Report

The report of Kiamehr et al. is very interesting. It deals with the compromised function of an induced pluripotent cell type and it is of interest of the scientific community at large. 

While I acknowledge the difficulty of such studies in general, I have to stress my major problem with the paper and that is the low number parallel experiments.

Authors report the derivation of 5 cell lines (in some cases 6 - page 2 lines 91-92 and page 6 line 101 page 4 line 114) of induced  pluripotent stem cells (iPSCs) differentiated into retinal pigment epithelium (RPE) cells. Three of these lines are from type 2 diabetic patients and two from healthy subjects. They compare the barrier function of these cells and state that iPSCs from diabetic patients have compromised barrier functions. These conclusions are drawn from three biological replicates in most cases (presented in Figs 1-5) but the results presented in Figs 6-8, if I underrstand clearly, were obtained on one cell line from both diabetic and healthy donors, only technical replicates were done. Although authors state this in the figure legends, it would be advisable to draw the attention of then readers to this fact in the body text (both in results and discussion). The best would be to have more biological replicates presented. 

Minor comments

1. I could not understand the description of the experiments with the tyrosinase gene (page 4 lines 134-136), or atleast it is not is synchrony with the figure description (page 5 line 152).

2. Incomprehensible sentence in oage 6 line 158-159. Please check.

3.It is hard to understand why the healthy control samples start to behave so much differently in the TEER experiment (Figure 3). DM, NG- and NGM- conditions are almost identical, if one comsiders the osmotic conditions. Could you explain this? The same applies for the cumulative permeabilty experiments (Fig. 4) where, to my opinion, samples in NG+ and NGM+ should behave the same way.

4. I would re-run the statistics or use a different test for the significant differences in Figure 9, Just by judging on the basis of the presented graphs, there might be a difference between stimulated conditions in c, d and f.    

Author Response

Response to reviewer #1 of manuscript number ijms-540733under the title “Compromised barrier function in human induced pluripotent stem cells -derived RPE cells from type 2 diabetic patients” by

Mostafa Kiamehr, Alexa Klettner, Elisabeth Richert, Ali Koskela, Arto Koistinen, Heli Skottman, Kai Kaarniranta, Katriina Aalto-Setälä and Kati Juuti-Uusitalo

The Reviewer #1

Authors report the derivation of 5 cell lines (in some cases 6 - page 2 lines 91-92 and page 6 line 101 page 4 line 114) of induced  pluripotent stem cells (iPSCs) differentiated into retinal pigment epithelium (RPE) cells. Three of these lines are from type 2 diabetic patients and two from healthy subjects. They compare the barrier function of these cells and state that iPSCs from diabetic patients have compromised barrier functions.

Our response: Thank you for noticing this. Originally, we had six hiPSC lines, but one line behaved unreliably thus that was omitted from the analyses. These are now corrected.

These conclusions are drawn from three biological replicates in most cases (presented in Figs 1-5) but the results presented in Figs 6-8, if I underrstand clearly, were obtained on one cell line from both diabetic and healthy donors, only technical replicates were done. Although authors state this in the figure legends, it would be advisable to draw the attention of then readers to this fact in the body text (both in results and discussion). The best would be to have more biological replicates presented.

Our response: We thank for this suggestion. We agree that there could have been more replicates in the results presented in Figs 6 – 8. To have more replicates, we would need to differentiate the hiPSC-RPE cells, and this together with new experiments would take in average 200 days. In this study that was 161 to 286 days. As advised, we have now clearly noted the number of experiments also in the body of the manuscript. These amendments can be found on page 9 on rows 236-237 and on row 250; on page 10 on row 279; on page 12 on rows 313 – 315; on page 14 on row 390 and on page 15 on rows 406,413 and 421 in the revised manuscript.

Minor comments

 I could not understand the description of the experiments with the tyrosinase gene (page 4 lines 134-136), or at least it is not is synchrony with the figure description (page 5 line 152).

Our response: This was a very good notion. We have now corrected the text to be in line with the Fig 2m and the figure legends. These can be now found on page 4, rows 134 – 136 in the revised manuscript.

Incomprehensible sentence in oage 6 line 158-159. Please check.

Our response: Thank you for noticing this. The sentence has now been separated to several shorter sentences, and edited to be hopefully more comprehensive. These can be now found on page 6, rows 164 – 165 in the revised manuscript.

It is hard to understand why the healthy control samples start to behave so much differently in the TEER experiment (Figure 3). DM, NG- and NGM- conditions are almost identical, if one comsiders the osmotic conditions. Could you explain this? The same applies for the cumulative permeabilty experiments (Fig. 4) where, to my opinion, samples in NG+ and NGM+ should behave the same way.

Our response: Yes, we agree with the esteemed referee. The similar phenomena can be seen also in the ECM gene expression where the NGM- and NG- had the biggest differences in both fibronectin and laminin A1 gene expression in healthy control hiPSCs in stimulated conditions. In most studies, either NG or NGM is used as a control. We do think it is an osmolality, which affects both to the maturation and gene expression. We have brought that up in the discussion on page 14 on rows 374 - – 376. It is known that osmolality affects to cellular functionality in RPE cells in several ways (Willermain F, Front. Physiol., 30 May 2014). However as we have not assessed that aspect by other methods, we wish to refrain from any further speculations in the discussion.

 I would re-run the statistics or use a different test for the significant differences in Figure 9, Just by judging on the basis of the presented graphs, there might be a difference between stimulated conditions in c, d and f.

Our response: Thank you for this suggestion. We re-run the statistics to all stimulated samples with all different combinations. We used two different nonparametric tests, for two samples, the Mann-Whitney U and the Kolmogorov–Smirnov, which were recommended by our statistician. The calculations revealed that even the samples, which appeared to have statistically significant different change in their gene expression remained above p = 0.05 with Mann-Whitney U and above p = 0.10 with Kolmogorov–Smirnov statistical tests. The p-values of samples, which from the graph might have been significantly altered, are listed below: In the type 2 diabetic hiPSC-RPEs stimulated with TNFa the difference in NGM- versus NG+ in collagen gene expression was only p = 0.05 with Mann-Whitney U and p = 0.1 with Kolmogorov–Smirnov, and for the same samples the fibronectin had p = 0.83 in Mann-Whitney U and p = 0.181 in Kolmogorov–Smirnov. Similarly, the healthy control hiPSC-RPEs stimulated with TNFa the biggest detected differences, which were seen the graph between NGM- and NG- in collagen gene expression, the p-value was 0.05 with Mann-Whitney U and 0.1 with Kolmogorov–Smirnov. In same healthy control hiPSC—RPEs stimulated with TNFa the fibronectin gene expression between NGM- versus NG+ was only p = 0.83 with Mann-Whitney U and p = 0.181 with Kolmogorov–Smirnov. Finally, in the healthy control hiPSC-RPEs stimulated with TNFa the biggest detected differences was between NGM- and NG- in laminin A1 gene expression, and the p-value was 0.05 with Mann-Whitney U and 0.1 with Kolmogorov–Smirnov. We do know that that this is mainly the result of low number of replicates. However, in our opinion it might bring some additional value to publish this data, which is replicated only once, as this line of study has already been terminated in our study group.

Reviewer 2 Report

The article of Kiamehr et al., is a straightforward study which investigates the effects of glucose concentration on cellular functionality of the induced pluripotent stem cell‐derived retinal pigment epithelial derived from type 2 diabetic and healthy control patients. 

The experimental design is well done, the results are clear and statistically valid, and the findings are novel.

I just have a few comments:

-A very limited part of the immunoblots is shown on the figure 7. Please, show larger part of the blots.

-Some references are missing. The authors should cite following papers: PMID: 28106278, PMID: 27486932 (page 14, line 349). 

Author Response

The Reviewer #2

A very limited part of the immunoblots is shown on the figure 7. Please, show larger part of the blots.

Our response: This was a good suggestion. We have now made a new figure 7 in which larger parts of blots are shown. In addition, we made amendment to the figure legend “The line on the left denotes the size of the blot.”

Some references are missing. The authors should cite following papers: PMID: 28106278, PMID: 27486932 (page 14, line 349).

Our response: The appearance of these publications were left unnoticed by us. Thank you for bringing these to our knowledge. The references are now added, and can be now found under the reference number 30. and 31. and on page 14 on row 348 in the revised manuscript.

Round 2

Reviewer 1 Report

A accapt the arguments and changes presented in the revised manuscript.